# Mortality Prediction Modeling for Patients with Breast Cancer Based on Explainable Machine Learning

**DOI:** 10.3390/cancers16223799

**Published:** 2024-11-12

**Authors:** Sang Won Park, Ye-Lin Park, Eun-Gyeong Lee, Heejung Chae, Phillip Park, Dong-Woo Choi, Yeon Ho Choi, Juyeon Hwang, Seohyun Ahn, Keunkyun Kim, Woo Jin Kim, Sun-Young Kong, So-Youn Jung, Hyun-Jin Kim

**Affiliations:** 1Department of Medical Informatics, School of Medicine, Kangwon National University, Chuncheon 24341, Republic of Korea; chicwon229@kangwon.ac.kr (S.W.P.);; 2Institute of Medical Science, School of Medicine, Kangwon National University, Chuncheon 24341, Republic of Korea; 3Cancer Data Center, National Cancer Control Institute, National Cancer Center, Goyang 10408, Republic of Korea; yelin@ncc.re.kr (Y.-L.P.);; 4Department of Surgery, Center of Breast Cancer, National Cancer Center, Goyang 10408, Republic of Korea; 5Department of Medical Oncology, Center for Breast Cancer, National Cancer Center, Goyang 10408, Republic of Korea; 6Department of Internal Medicine, Kangwon National University Hospital, Chuncheon 24289, Republic of Korea; 7Department of Internal Medicine, School of Medicine, Kangwon National University, Chuncheon 24341, Republic of Korea; 8Targeted Therapy Branch, Research Institute, National Cancer Center, Goyang 10408, Republic of Korea; 9Department of Laboratory Medicine, Hospital, National Cancer Center, Goyang 10408, Republic of Korea

**Keywords:** breast cancer, artificial intelligence, machine learning, explainable artificial intelligence, mortality

## Abstract

Breast cancer is the most common cancer in women worldwide, and strategic efforts are required to reduce its mortality. Past studies have usually focused on a limited number of clinical or demographic factors to predict breast cancer prognosis. However, we used thirty-one features, including demographic characteristics, laboratory results, pathology, and treatment information, to predict breast cancer mortality. In addition, the Shapley Additive Explanation (SHAP) method, an explainable artificial intelligence technique, was used. This approach allows us to identify and interpret the key features that have a significant impact on breast cancer mortality. Key predictors of the mortality classification model included occurrence in other organs, age at diagnosis, N stage, T stage, curative radiation treatment, and Ki-67(%). Accurate breast cancer mortality prediction and detection of risk factors based on machine learning may provide opportunities for appropriate therapeutic interventions such as early chemotherapy, surgery, and other measures that may reduce mortality.

## 1. Introduction

Breast cancer in women is a major public health problem with the highest incidence among cancers and the highest cancer disability-adjusted life-years [1,2]. According to the World Health Organization (WHO), over 2.3 million women were diagnosed with breast cancer in 2020, resulting in 685,000 deaths [3]. The mortality rate of breast cancer is expected to increase, reaching 1 million worldwide by 2040 [4]. Therefore, at the time of initial diagnosis, it is very important to identify the risk factors related to recurrence or survival for each patient and to provide appropriate management plans and treatments, such as surgery, radiation therapy, and chemotherapy. Therefore, predicting mortality and identifying risk factors for breast cancer patients are essential for improving clinical decision-making, controlling the patient’s environment, and enabling appropriate interventions. At the time of initial diagnosis, identifying risk factors related to recurrence or survival for each patient is crucial, allowing for tailored management plans and treatments, such as surgery, radiation therapy, and chemotherapy [5,6].

Traditionally, statistical analyses have been used to investigate breast cancer mortality rates. Recently, novel machine learning (ML) models have demonstrated superior predictive capabilities compared to traditional methods such as logistic regression or Cox regression analysis [7,8,9]. Several studies have demonstrated that ML models have good predictive abilities in identifying patients with breast cancer [10,11,12,13]. Many previous studies have mainly focused on considering only one risk factor, such as BRCA1/2 pathogenic variants, neo-adjuvant chemotherapy, or mammography, to predict breast cancer prognosis [5,14,15]. However, considering the differences in individual mortality rates, even for the same type of cancer, it is important to predict the prognosis using various clinical characteristics for personalized treatment [16,17,18,19,20]. For this reason, more recent studies have reported models for predicting personalized prognosis using various clinical characteristics [17,18,19,20,21,22,23]. In addition, many studies using large cohort datasets such as the Surveillance, Epidemiology, and End Results Program (SEER) [24,25,26,27,28] and the National Cancer Data Base (NCDB) 10–14 [29,30,31] have been instrumental in studying predicting prognoses and proposing treatment methods for breast cancer. These studies have demonstrated risk variables by integrating data on demographic information with clinical and pathologic factors in breast cancer patients. Despite many research advancements, applying findings from Western datasets to the Korean population is limited by genetic, environmental, and other differences, requiring cautious generalization.

Thus, in this study, we developed predictive classification models for breast cancer mortality, identified risk factors, and confirmed the most optimized ML model by using real-world data. We used 31 risk factors, including occurrence in other organs, N stage, chemotherapy, histologic grade, and tumor subtype, to predict breast cancer survival rates.

In addition, we used the SHapley Additive exPlanation (SHAP) method, an explainable artificial intelligence (XAI) technique, for more detailed model interpretation. Consequently, we can overcome the limitations of ML characteristics, which are black-box models, and discover and interpret key features that significantly affect breast cancer mortality (Figure 1).

## 2. Materials and Methods

### 2.1. Materials

#### 2.1.1. Study Population and Data Collection

This retrospective study analyzed data from patients diagnosed with breast cancer between January 2005 and December 2020, sourced from the National Cancer Center (NCC), Korea database, a prospectively collected archive of breast cancer treatments at the NCC [32]. Mortality data, including causes of death until 2021, were obtained from Statistics Korea and linked to the NCC database to identify survival status. Initially, 13,165 patients were screened for eligibility, as shown in Figure 2. Given that our study focused on female patients diagnosed with breast cancer, we excluded 1879 patients based on the following criteria: male patients (N = 1), patients without a breast cancer disease code (N = 1), foreign patients (N = 166), patients diagnosed with other cancers before breast cancer (N = 324), patients with benign tumors (N = 858), patients with missing information on histologic grade and TNM stage (N = 167), and patients receiving radiation therapy for palliative treatment (N = 362). After these exclusions, 11,286 patients with a first diagnosis of breast cancer remained, with 695 deaths (6%) and 10,591 survivors (94%). We analyzed the data and found that the occurrence of cancer in other organs after breast cancer diagnosis had the greatest impact on predicting breast cancer mortality. Accordingly, this study was divided into two groups to understand the effect of other variables on mortality prediction. One group included all patients with breast cancer, while the other excluded patients who developed other cancers after a breast cancer diagnosis (n = 1071), leaving a total of 10,215 patients. Among them, 426 (4%) died and 9789 (96%) survived. These were calculated using covariates such as age at diagnosis, height, alcohol consumption, p53(%), Ki-67(%), and human epidermal growth factor receptor 2 (HER2).

#### 2.1.2. Variables

Of the 64 raw data features constructed in the NCC registry shown in Appendix A, 33 were excluded. Date-related variables were deemed unnecessary for this study’s model implementation (N = 7), and there were a large number of missing values and data imbalances between groups (N = 10). Furthermore, variables were excluded to prevent multicollinearity based on clinical experts’ opinions and data (N = 16). Thus, we used 31 features in five main groups, including patient’s health and demographic information, laboratory results, treatment types, pathology, and other variables, as summarized in Appendix A. Operative features refer to the types of surgical procedures performed for cancer removal, such as breast-conserving surgery (BCS) and mastectomy. Radiotherapy treatment status was determined based on its use for purely curative purposes. Chemotherapy was classified as adjuvant, neoadjuvant, or none. Additional treatments were categorized based on hormone levels and targeted therapy. Experienced breast surgeons from the NCC Pathology Department evaluated the pathological data, which included tumor location (bilateral or unilateral) and type (synchronous or metachronous). Tumor subtypes were classified by hormone receptor and HER2 status as follows: Luminal A (ER+ and/or PR+, and HER2−), Luminal B (ER+ and/or PR+, and HER2+), HER2-overexpressing (ER− and PR−, and HER2+), and basal-like (ER− and PR−, and HER2−). The p53(%) and Ki-67(%) molecular pathology results were used as prognostic markers for early breast cancer to determine the need for further adjuvant chemotherapy. TNM stage was assigned according to the American Joint Committee on Cancer (AJCC) Cancer Staging Manual, 7th edition. The T and N stages, not described in the TNM stage classification, were defined under the guidance of an experienced clinician according to the AJCC criteria, using the size of the primary tumor and the number of lymph nodes.

### 2.2. Methods

#### 2.2.1. Data Pre-Processing

Prior knowledge about raw data can significantly impact the performance of optimized classifiers, highlighting the importance of data pre-processing in effective mortality classification using machine learning algorithms. The dataset used in this study comprised clinical patient treatment records, where a few outliers—resulting from incorrect entries by medical staff in the electronic medical records (EMR)—were removed. Missing values for features such as age at diagnosis, height, alcohol consumption, smoking status, age at menarche and menopause, and family history were imputed. Continuous variables were imputed with means, while categorical variables were imputed using their respective modes. This approach ensured consistent and efficient handling of missing data. Although mean and mode imputation may not capture all data complexities, they offer a straightforward and effective method for minimizing the risk of introducing additional biases. The number of cases where imputation was performed for each feature is detailed in Appendix A. Given the relatively high missing data rate, we selected the mean for continuous variables and the mode for categorical variables [33,34,35]. Continuous variables such as height (mean = 157.35), age at menarche (mean = 14.79), and age at menopause (mean = 49.51) were imputed with their respective means. Categorical variables such as alcohol consumption (mode = 0 [No]), smoking status (mode = 0 [No]), and family history (mode = 0 [No]) were imputed with their respective modes. Mode-based imputation, which fills missing values with the most frequent category, maintains consistency when a majority of observations share a common feature. Sensitivity analyses showed that this strategy did not adversely impact model performance. Furthermore, data from pathology reports were used to impute T- and N-stage data. The T stage was supplemented with pathologic stage and tumor size, while the N stage was supplemented with pathologic stage information. Additionally, skewness and kurtosis methods were employed to identify variables with biased distributions and potential outliers. Log transformation was applied to fasting glucose levels and white blood cell counts, followed by normality tests for each variable. Finally, min-max normalization was conducted on all input variables to prevent issues arising from differences in data scale, enhancing model performance. This normalization step was crucial, as features with larger scales could disproportionately influence the machine learning model compared to those with smaller values [34].

#### 2.2.2. Machine Learning

Four ML models, such as LR, random forest (RF), extreme gradient boosting (XGB), and light gradient boosting machine (LGB), were constructed to classify mortality in breast cancer patients. LR is a traditional probabilistic statistical model widely used in the medical sciences [35]. RF is an ensemble method in ML that constructs numerous decision trees during training and outputs a class, with the final classification determined by the mode or mean of individual tree outcomes [36]. XGB is a tree-based ensemble ML algorithm that combines multiple decision trees and employs classification and regression techniques based on gradient descent. It expands decision trees horizontally (i.e., level-wise) to reduce their depth and works well on imbalanced datasets with excellent accuracy and speed [37]. LGB is an algorithm similar to XGB but can learn faster on large datasets [38]. These classifiers were trained using 31 features, including demographic characteristics, laboratory results, pathology, and treatment information. Mortality prediction was based on the presence or absence of other carcinomas after breast cancer diagnosis. All ML programming for this study was performed using Python (version 3.8.10), with the models constructed using Scikit-learn (version 1.2.0). Hyperparameter tuning was performed using Optuna (version 3.3.0), a hyperparameter optimization framework.

#### 2.2.3. Model Algorithm Development and Evaluation

Of the total data, 80% were allocated to training, while the remaining 20% were used for testing. Validation data were used for 20% of the total training set. For each of the four classification methods, 20% of the selected participant data were completely removed from the cross-validation (CV)-based hyper-parameter value estimates. Stratified k-fold CV (k = 5) was conducted to avoid label distortions during model generation and to maintain stability. The performance measures included accuracy, precision, recall, F1 score, area under the receiver operating characteristic curve (AUC), specificity, Brier score, Matthews correlation coefficient (MCC), and area under the precision–recall curve (AUPRC) [39,40,41,42]. Recall was calculated as true positive (TP)/(TP + false negative (FN)) and specificity as true negative (TN)/(TN + false positive (FP)). The F1 score is the harmonic mean of precision and recall, where precision equals TP/(TP + FP). These criteria are commonly used to report model evaluations [43,44,45]. Additionally, the Brier score offers a direct measure of the accuracy of probabilistic predictions by assessing both calibration and sharpness. It ranges from 0 to 1, with lower values indicating a model that produces more precise predictions [46]. The MCC provides a balanced metric that accounts for true and false positives and negatives, calculated using the formula (TP × TN − FP × FN)/((TP + FP)(TP + FN)(TN + FP)(TN + FN))^1/2^. The AUPRC emphasizes performance in imbalanced datasets by focusing on precision and recall. In addition, model classification predictions were elucidated using SHAP, a model explanation method based on feature importance. SHAP utilizes cooperative game theory to determine how each feature contributes to ML model predictions, allowing for the interpretation of model performance [47,48]. SHAP assumes independence between features, and correlated features could potentially affect the reliability of SHAP values. To address this, we analyzed the correlations among features used in the model. A correlation heatmap was generated to visually inspect feature relationships, showing that the features were not highly correlated with each other, allowing us to reasonably assume independence. The correlation heatmap is provided in the Appendix A (Appendix A). We used Pearson correlation coefficients and observed that no pairs of features exceeded a correlation threshold of 0.7, indicating minimal multicollinearity.

#### 2.2.4. Statistical Analysis

To enhance the robustness and generalizability of our primary findings and balance the baseline covariates, we employed an exposure-driven 1:3 propensity score matching (PSM) analysis while minimizing a logistic regression model with the implications of potential confounders [49]. The quality of the match was evaluated through the standardized mean difference (SMD), with an SMD < 0.1 indicating a negligible difference between the groups (Appendix A) [50]. All patient characteristics are presented as means with standard deviations or counts (%). An independent *t*-test was performed to assess the differences in the mean between the alive and death groups for continuous variables. Categorical variables are compared between two groups using Chi-squared (χ^2^) tests. Considering the multiple testing of the same cohort, we used a Bonferroni correction based on the number of tested samples in the same cohort [significance threshold of α = 0.05/2(number of tested samples in the same cohort) = 0.025]. In addition, the Cox proportional hazards regression model was fitted after verifying the proportional hazards assumption using the log-log plot (Appendix A). We assessed the hazard ratio (HR) of all variables, using univariate and multivariate Cox regression analysis. We selected the high-impact variables identified consistently across the four ML models and used them in Kaplan–Meier (KM) analysis to assess their effects on survival. For survivors, the duration was calculated from the initial breast cancer diagnosis to the end of the observation period, while for those who passed away, it was calculated from the diagnosis to the date of death. The KM analysis provided insights into survival rates over the entire observation period, tracking changes in breast cancer patient survival. To determine the statistical significance of the top variables identified by SHAP, we performed KM analysis, allowing us to both visually and statistically evaluate the influence of each variable on survival and mortality. The analysis yielded significant *p*-values for each variable [51], with all statistical tests conducted at a significance level of *p* < 0.001. All analyses were performed using R software (version 3.6.3).

## 3. Results

### 3.1. Patient Characteristics

In this study, we developed predictive ML classification models to predict mortality in patients with breast cancer. We formed two groups: (1) all patients with breast cancer (overall breast cancer group) and (2) excluding patients who developed other cancers after breast cancer (breast-cancer-only group). The first group comprised 11,286 patients (10,591 alive and 695 deceased), and the second group included 10,215 patients (9789 alive and 426 deceased), as shown in Appendix A. Using 1:3 PSM, we analyzed 2780 cases (2085 alive and 695 deceased) in the overall breast cancer group and 1704 cases (1278 alive and 426 deceased) in the breast-cancer-only group.

Table 1 presents the characteristics of the two groups. In the overall group, the incidence of cancer in areas other than the breast was 38.7% (*p* < 0.001) among the deceased, which was approximately five times higher than that among the survivors. The T stage (*p* < 0.001) and N stage (*p* < 0.001) showed higher rates among the deceased as the disease progressed. For tumor subtypes (*p* < 0.05), Luminal A (63.0%), basal (16.1%), Luminal B (13.8%), and HER2 overexpression (7.1%) were observed among the deceased. Additionally, 78.8% (*p* < 0.001) of survivors received adjuvant chemotherapy at a higher rate than the deceased. In the second group, T stages 2–4 and N stages 1–3 showed higher rates among the deceased as the disease progressed compared to the survivors. Adjuvant chemotherapy was administered to 78.7% (*p* < 0.001) of survivors, which was higher than that of the deceased. Furthermore, the rate of hormone treatment (*p* < 0.001) among the survivors was high at 79.2%.

### 3.2. Cox Proportional Hazard Model

Table 2 presents the results of the univariate and multivariate analyses. In the univariable results for overall breast cancer, age at diagnosis was associated with the outcome (HR 1.03, 95% confidence incidence (CI) [1.03, 1.04]). Other significant predictors included height (HR 0.96, 95% CI [0.95, 0.97]), age at menarche (HR 1.08, 95% CI [1.03, 1.13]), fasting glucose (HR 1.01, 95% CI [1.00, 1.01]), and occurrence in other organs (HR 3.84, 95% CI [3.30, 4.48]). For the breast-cancer-only group, age at diagnosis was also associated with the outcome (HR 1.05, 95% CI [1.04, 1.06]). Other significant predictors included height (HR 0.94, 95% CI [0.92, 0.96]), age at menarche (HR 1.17, 95% CI [1.09, 1.26]), fasting glucose (HR 1.00, 95% CI [1.00, 1.01]), and WBC (HR 1.07, 95% CI [1.03, 1.11]).

In the multivariable results for overall breast cancer, age at diagnosis was associated with the outcome (HR 1.03, 95% CI [1.02, 1.04]). Other significant predictors included height (HR 0.99, 95% CI [0.97, 1.00]), age at menarche (HR 0.96, 95% CI [0.91, 1.02]), fasting glucose (HR 1.00, 95% CI [1.00, 1.00]), and occurrence in other organs (HR 2.57, 95% CI [2.17, 3.04]). For the breast-cancer-only group, age at diagnosis was also associated with the outcome (HR 1.04, 95% CI [1.03, 1.05]). Other significant predictors included height (HR 1.00, 95% CI [0.98, 1.02]), age at menarche (HR 1.08, 95% CI [1.01, 1.14]), fasting glucose (HR 1.00, 95% CI [1.00, 1.01]), and WBC (HR 1.01, 95% CI [0.97, 1.05]).

### 3.3. Model Performance and Evaluation

Four ML algorithms (LR, RF, XGB, and LGB) were used to construct predictive models for breast cancer mortality. The results for both groups are presented separately. The model was trained using the 31 finally selected features (Appendix A). Model performance is presented in Table 3.

In the overall breast cancer group, XGB exhibited the highest discriminative ability, with an AUC of 0.8722, an F1 score of 0.6121, and an AUPRC of 0.7130, followed by LGB with an AUC of 0.8677, an F1 score of 0.6345, and an AUPRC of 0.7059, as shown in Figure 3A. XGB also had the highest accuracy of 0.8381 among the four ML models, followed by LGB with an accuracy of 0.8094 and RF with an accuracy of 0.8058.

In the breast-cancer-only group, XGB showed the highest discriminative ability, with an AUC of 0.8518, an F1 score of 0.6107, and an AUPRC of 0.7013, followed by LGB with an AUC of 0.8482 and an accuracy of 0.8270, as shown in Figure 3B. In addition, XGB demonstrated the highest accuracy of 0.8504, followed by LGB with an accuracy of 0.8270 and RF with 0.7859. 

However, although XGB presents the best performance with regards to its AUC, accuracy, and AUPRC, when considering model performance for imbalanced data and minority classes in both groups, it does not show the best performance, with an F1 score lower than LGB for both groups.

### 3.4. Feature Importance and Interpretation

SHAP values indicate the contributions of individual variables to the predictive classification model results. They help interpret the influence and importance of each feature in the model’s decision-making process. The x-axis represents the SHAP value, which indicates the effect of a feature on the prediction outcome, and the y-axis lists the features. Figure 4 illustrates the XAI results for XGB for each group. The other models’ results are presented in Appendix A. The SHAP summary plot provides an information-dense summary of the effects of the top features of a dataset on the model results (Figure 4). For each instance, the explanation is represented by a single dot for each feature flow. The distribution of points on the plot for each variable across the SHAP value axis indicates the degree of the impact of this feature on the output of the model. Each point represents an individual data point in the dataset.

In the overall breast cancer group, variables such as occurrence in other organs, age at diagnosis, N stage, radiation treatment for curative purposes, height, Ki-67(%), and T stage had a significant impact on the XGB model, which exhibited the highest performance among all variables.

The top four variables, occurrence in other organs, N stage, age at diagnosis, and radiation treatment for curative purposes, were found to have a significant influence on individual model results, which was consistent across the other models as well. The interpretation of SHAP values suggests that the occurrence of other cancers after breast cancer, older age at diagnosis, higher N stage, performance of radiation treatment for curative purposes, and higher Ki-67(%) values are more important in determining the model results. These features are meaningful indicators of mortality predictions.

Similarly, in the breast-cancer-only group, variables such as N stage, age at diagnosis, T stage, p53(%), and Ki-67(%) had a significant impact on the XGB model, which exhibited the highest accuracy among all variables. The variables of T and N stages, age at diagnosis, and Ki-67(%) were found to have a significant influence on the individual model results, consistent with other models. In particular, p53(%) and WBC were included as important variables, compared to the overall breast cancer group.

SHAP analysis revealed that certain high-impact variables significantly contribute to the model’s predictions, indicating their influence on patient outcomes. Specifically, these findings suggest a positive correlation between these variables and survival rates, implying that an increase in their frequency or grade may positively affect survival outcomes. Appendix A shows the KM survival curves for high-impact variables in the ML models applied to the overall breast cancer group. Furthermore, Appendix A shows the KM survival curves for the high-impact variables in the ML models applied to the breast-cancer-only group.

## 4. Discussion

In this study, we developed and evaluated ML models to predict mortality in patients with breast cancer using a comprehensive dataset from the NCC in Korea. Our findings demonstrate the efficacy of ML-based predictive models in accurately classifying patient mortality and highlight the importance of XAI in enhancing the interpretability of these models for clinical decision support systems. Breast cancer remains a significant public health challenge with high incidence rates globally [1,2,3]. Accurate prediction of mortality in patients with breast cancer is crucial for timely intervention and personalized treatment planning. Traditional methods, such as LR and Cox regression analysis, have been widely used but often fall short in terms of predictive performance. Our study showed that advanced ML techniques, including XGB and LGB, outperform traditional methods, providing higher accuracy and specificity for mortality prediction. In our study, the results indicate that the XGB model exhibited the highest discriminative ability, with an AUC of 0.8722 for the overall breast cancer group and 0.8518 for the breast-cancer-only group. The LGB model also showed robust performance, particularly in the second group, with an accuracy of 0.8270. Compared with the existing literature, we utilized advanced ML models and comprehensive data. Hou et al. [52], Nguyen et al. [17], and Allugunti [53] used advanced ML models to improve predictive survival performance. Similar to these studies, we conducted predictive classification for the mortality of patients with breast cancer using advanced models, such as XGB and LGB, achieving high predictive accuracy. A notable strength of our study is the feature selection and importance analysis. Similar to previous studies [54], we emphasized feature importance analysis using SHAP values, providing deeper insights into how individual variables influence mortality predictions with high accuracy. Additionally, KM analysis was used to confirm the survival rates for each category of high-impact variables identified by the ML models. An essential aspect of our study is the application of SHAP values to interpret the model predictions. SHAP values provide insights into the contribution of individual variables, such as age at diagnosis, N stage, T stage, and treatment modalities (e.g., radiation and chemotherapy), to the model’s decision-making process [55]. Interestingly, height was included as an important variable, which may somewhat support the association between height and breast cancer risk reported in previous studies [56,57]. In particular, p53 (%) and WBC were identified as important variables in the breast-cancer-only group. In general, somatic mutations in the p53 gene are common in triple-negative breast cancer with a poor prognosis. However, since the p53 used in this study was based on immunohistochemistry results, it was difficult to identify the association with gene expression or mutation. Further studies are needed to prove the link between them. In addition, our WBC results suggest that additional studies may be needed to identify the relationship between the neutrophil/lymphocyte ratio and poor prognoses. This level of interpretability is crucial for clinicians to understand the factors driving mortality predictions and make informed decisions regarding subject management.

This study had several strengths and unique contributions. First, we used PSM to create a balanced dataset to ensure robust model training and evaluation [58,59]. PSM helped minimize the bias commonly associated with observational studies and improved the reliability of our model predictions. Rajendran et al. [59] addressed class imbalance in breast cancer prediction using techniques such as the synthetic minority oversampling technique [60], under-sampling [61], and hybrid methods. However, although these data augmentation and reduction methods can increase the learning effect of the model, they can be detrimental to the error values that exist in real data, such as outliers. Therefore, we applied the PSM method to ensure that data features and labels were adequately balanced [62,63,64]. Unlike simple oversampling or under-sampling techniques, PSM allowed us to match patients based on key covariates, minimizing selection bias and creating a realistic dataset for training. This approach handled real-world variability more effectively and improved model generalizability by reducing class overrepresentation, resulting in a more balanced view of patient outcomes. In summary, PSM improved model performance while mitigating biases and maintaining data integrity, leading to more reliable predictions, especially in observational datasets.

Second, unlike other studies that primarily focused on predictive performance, our study emphasized the interpretability of model predictions using SHAP values [65]. This approach provides a transparent view of how each feature influences the model’s decisions, which is essential for clinical applications [58]. SHAP values offer a detailed explanation by considering the offsets among all variables and using permutation calculations. This approach ensures that our ML models are accurate and interpretable, making them practical for real-world clinical implementation. Third, we used a dataset that included a wide range of features, including demographic, clinical, pathological, and treatment-related variables, ensuring a holistic approach to mortality prediction. Fourth, our study utilizes real-world data from the NCC, providing a realistic and practical basis for model development and evaluation. Using real-world clinical data offers several advantages, including greater diversity in subject demographics, treatment protocols, and disease progression patterns. In addition, real-world data reflect actual clinical conditions and variations, making the models more applicable and reliable for routine clinical decision-making. This enhances the applicability and relevance of our findings to clinical practice.

However, this study had a few limitations. First, the retrospective nature of this study and data from a single NCC may limit the generalizability of our findings. Moreover, the imbalance between survivors and deaths could introduce bias, although we addressed this issue using PSM. To the best of our knowledge, there is currently no established standard for the optimal ratio of PSM, as long as the case-to-control ratio is not too large (e.g., 1:5 or more). In general, PSM has been performed with a 1:1 ratio, which is also the default value for most PSM tools. However, in this study, the number of deaths among breast cancer patients was relatively small (n = 695, 6% of the total); therefore, we thought that the number of samples included in the machine learning model would be very small if PSM is performed with a ratio of 1:1. For this reason, we considered a 1:3 PSM ratio, referring to previous studies that used a 1:3 PSM ratio for analyses related to the survival of breast cancer patients [66,67,68]. We used approximately 20% of the total data for ML training. Future studies should consider a multimodal approach that combines structural and nonstructural data, such as biosignals and images. Second, this study was cross-sectional, with a focus on mortality prediction and without longitudinal data. Longitudinal studies are needed to design more precise and personalized treatments and validate the utility of these models in continuous subject monitoring. Second, we were unable to perform an external validation of our models, which is crucial for confirming the generalizability and robustness of our findings across different populations. Third, this study included a large number of imputed data using the mean or mode of the variables due to the high rate of missing variables in the EMR data. In addition to the large amount of missing data, this simple imputation approach for missing data could potentially be a major limitation of this study. Recently, various statistical inference methods for missing data (e.g., inference based on maximum likelihood estimates) have been reported, and further research is needed on how actual model results differ between various imputation approaches. In addition, future studies need to make additional efforts to reduce the rate of missing data in datasets. Fourth, the recall of the XGB model was notably lower compared to other performance metrics. Recall was calculated as TP/(TP + FN). In other words, recall is the ratio of the correctly predicted positive class to all classes. In imbalanced datasets, where the positive class (1:death) is often the minority, models tend to predict most data points as belonging to the majority class (0:alive). This leads to an increase in false negatives, as the model misses a significant portion of the minority class. As a result, the sensitivity (or recall) tends to decrease. Therefore, in imbalanced datasets, models are more likely to overlook positive cases, causing sensitivity to drop. To address this limitation, we introduced the Area Under the Precision–Recall Curve (AUPRC) as an additional performance metric. The AUPRC can effectively complement the F1 score, as it evaluates the model’s precision and recall performance across all thresholds rather than just one. This additional measure is particularly useful in imbalanced datasets, where understanding model performance across a range of decision thresholds can reveal nuances that the single-point F1 score may miss. In future studies, to improve this limitation, techniques such as class weighting or using loss functions that focus on recall (such as focal loss) will be used [69]. Fifth, interpretation through SHAP has potential issues because the attribution of feature importance is typically based on random permutations. High correlations among the predictors can lead to SHAP values that may not accurately represent the importance of each feature. In future studies, we plan to implement Kernel SHAP, a method that employs a weighted sampling strategy to calculate each SHAP value, considering the intercorrelations among features. Finally, we used data classified based on whether cancer had occurred in other organs after breast cancer diagnosis to understand its impact on survivors. However, we could not definitively determine whether the malignancies in other organs were new cancers that developed after the breast cancer diagnosis or metastases from the original breast cancer. Future studies are required to establish more precise criteria for patient classification to ensure a clearer distinction between metastatic diseases and new primary malignancies. Given the current data limitations, we included patients with the involvement of other organs; however, future work should aim to more meticulously exclude these cases to differentiate between metastasis and new cancer development accurately.

## 5. Conclusions

This study demonstrated the predictive classification of breast cancer mortality using cause of death information from Statistics Korea. By utilizing real-world data, such as clinical diagnosis and treatment information, our results are more reliable. We constructed four ML models and achieved high accuracy and AUC using 31 risk factors for breast cancer. Furthermore, through XAI, we identified key variables affecting breast cancer mortality, such as occurrence in other organs, age at diagnosis, N stage, and T stage. Our study results may help physicians determine treatment for the prognostic management of breast cancer patients by predicting the prognosis related to survival. However, a multi-center extension study on representative large-scale data is needed to reduce missing values in EMR data and to develop a more powerful model.

## Figures and Tables

**Figure 1 cancers-16-03799-f001:**
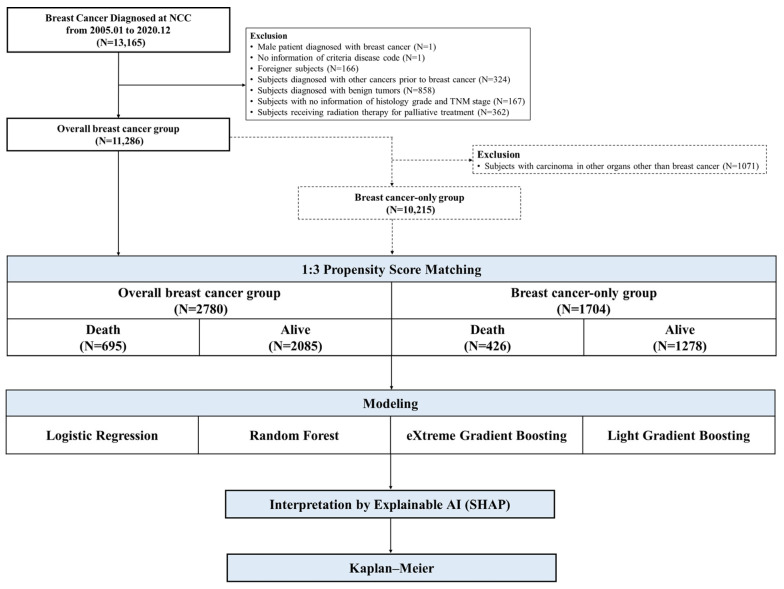
The scheme of the study flow.

**Figure 2 cancers-16-03799-f002:**
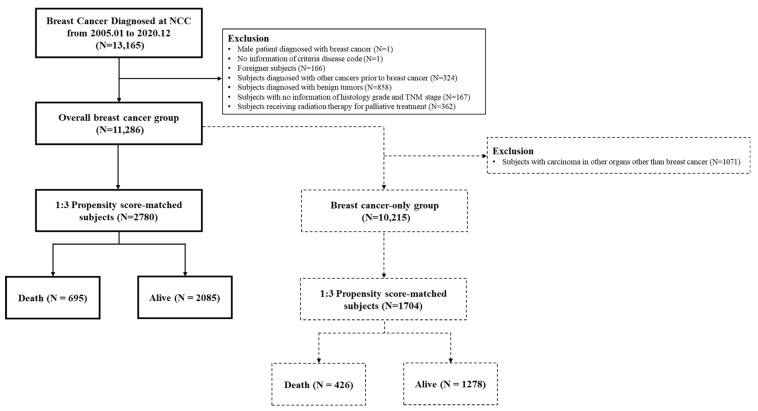
Flow chart of clinical variables with participants.

**Figure 3 cancers-16-03799-f003:**
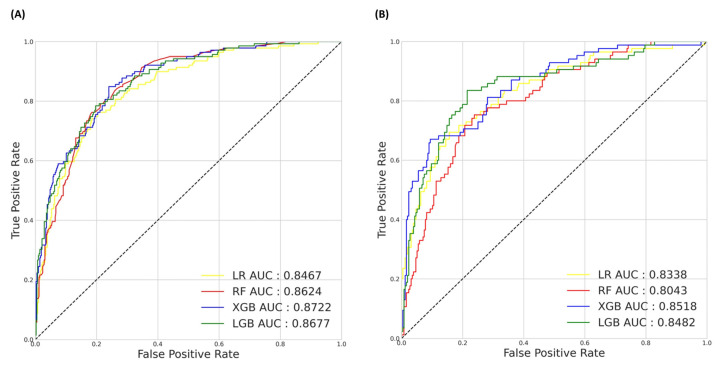
(**A**) ROC curve for the overall breast cancer group, (**B**) ROC curve for the breast-cancer-only group. ROC, Receiver operating characteristic.

**Figure 4 cancers-16-03799-f004:**
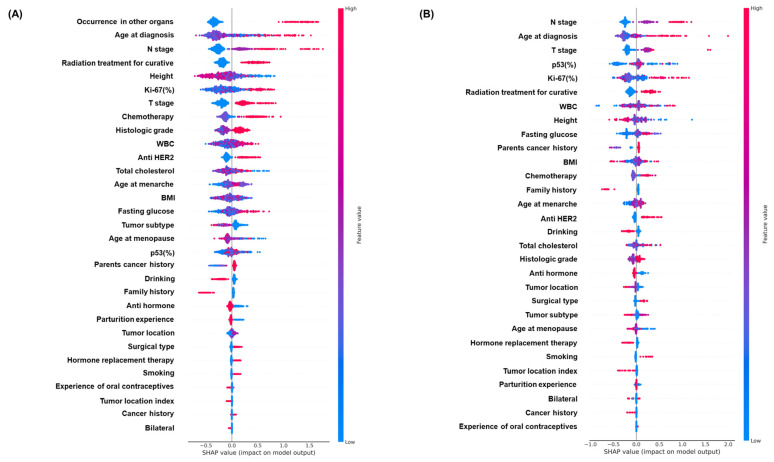
The SHAP summary plots of the XGB: (**A**) overall breast cancer group and (**B**) breast-cancer-only group. SHAP, Shapley additive explanations; XGB, extreme gradient boosting.

**Table 1 cancers-16-03799-t001:** Description of participant attributes after propensity score matching.

	Overall Breast Cancer Group				Breast-Cancer-Only Group			
	Total (N = 2780)	Death (N = 695)	Alive (N = 2085)	*p*	Total (N = 1704)	Death (N = 426)	Alive (N = 1278)	*p*
**Age at diagnosis**	51.4 ± 11.4	54.7 ± 14.0	50.3 ± 10.1	<0.001	52.1 ± 12.0	57.0 ± 14.6	50.5 ± 10.5	<0.001
**Height**	157.1 ± 5.8	155.6 ± 6.2	157.5 ± 5.6	<0.001	156.8 ± 6.0	155.2 ± 6.3	157.4 ± 5.8	<0.001
**BMI**	23.9 ± 3.5	24.3 ± 3.6	23.8 ± 3.5	<0.05	23.9 ± 3.6	24.4 ± 3.7	23.8 ± 3.6	<0.05
**Smoking**				0.058				0.518
No	2605 (93.7)	638 (91.8)	1967 (94.3)		1607 (94.3)	397 (93.2)	1210 (94.7)	
Yes	175 (6.3)	57 (8.2)	118 (5.7)		97 (5.7)	29 (6.8)	68 (5.3)	
**Drinking**				<0.001				<0.001
No	2214 (79.6)	600 (86.3)	1614 (77.4)		1335 (78.3)	373 (87.6)	962 (75.3)	
Yes	566 (20.4)	95 (13.7)	471 (22.6)		369 (21.7)	53 (12.4)	316 (24.7)	
**Age at menarche**	14.9 ± 1.5	15.1 ± 1.5	14.8 ± 1.5	<0.001	14.9 ± 1.6	15.2 ± 1.5	14.8 ± 1.5	<0.001
**Age at menopause**	49.5 ± 3.3	49.3 ± 3.8	49.5 ± 3.1	0.160	49.5 ± 3.5	49.3 ± 4.0	49.5 ± 3.3	0.566
**Parturition experience**				0.907				0.969
No	327 (11.8)	85 (12.2)	242 (11.6)		218 (12.8)	53 (12.4)	165 (12.9)	
Yes	2453 (88.2)	610 (87.8)	1843 (88.4)		1486 (87.2)	373 (87.6)	1113 (87.1)	
**Experience of oral contraceptives**				0.522				0.963
No	2507 (90.2)	619 (89.1)	1888 (90.6)		1526 (89.6)	380 (89.2)	1146 (89.7)	
Yes	273 (9.8)	76 (10.9)	197 (9.4)		178 (10.4)	46 (10.8)	132 (10.3)	
**Hormone Replacement Therapy**				0.399				0.821
No	1943 (93.2)	637 (91.7)	2580 (92.8)		1547 (90.8)	390 (91.5)	1157 (90.5)	
Yes	142 (6.8)	58 (8.3)	200 (7.2)		157 (9.2)	36 (8.5)	121 (9.5)	
**Family history**				<0.05				<0.001
No	2587 (93.1)	667 (96.0)	1920 (92.1)		1575 (92.4)	414 (97.2)	1161 (90.8)	
Yes	193 (6.9)	28 (4.0)	165 (7.9)		129 (7.6)	12 (2.8)	117 (9.2)	
**Parents’ cancer history**				<0.001				<0.001
Paternity	273 (9.8)	40 (5.8)	233 (11.2)		145 (8.5)	23 (5.4)	122 (9.5)	
Maternal line	158 (5.7)	16 (2.3)	142 (6.8)		124 (7.3)	10 (2.3)	114 (8.9)	
Parental	53 (1.9)	6 (0.9)	47 (2.3)		32 (1.9)	3 (0.7)	29 (2.3)	
None	2296 (82.6)	633 (91.1)	1663 (79.8)		1403 (82.3)	390 (91.5)	1013 (79.3)	
**Cancer history**				0.986				0.884
No	2665 (95.9)	667 (96.0)	1998 (95.8)		1654 (97.1)	415 (97.4)	1239 (96.9)	
Yes	115 (4.1)	28 (4.0)	87 (4.2)		50 (2.9)	11 (2.6)	39 (3.1)	
**Total cholesterol**	194.8 ± 36.6	194.3 ± 39.2	195.0 ± 35.8	0.907	195.0 ± 36.7	196.4 ± 40.2	194.5 ± 35.5	0.671
**Fasting glucose**	113.2 ± 41.6	122.8 ± 54.4	110.0 ± 35.7	<0.001	113.7 ± 40.6	124.2 ± 54.6	110.2 ± 34.0	<0.001
**WBC**	6.5 ± 2.2	6.8 ± 2.5	6.3 ± 2.1	<0.001	6.5 ± 2.3	6.9 ± 2.8	6.4 ± 2.1	<0.05
**Surgical type**				<0.001				<0.001
None	40 (1.4)	9 (1.3)	31 (1.5)		23 (1.3)	5 (1.2)	18 (1.4)	
BCS	2395 (86.2)	529 (76.1)	1866 (89.5)		1450 (85.1)	322 (75.6)	1128 (88.3)	
Mastectomy	345 (12.4)	157 (22.6)	188 (9.0)		231 (13.6)	99 (23.2)	132 (10.3)	
**Tumor location**				0.920				0.330
Left	1359 (48.9)	350 (50.4)	1009 (48.4)		833 (48.9)	224 (52.6)	609 (47.7)	
Right	1296 (46.6)	313 (45.0)	983 (47.1)		796 (46.7)	189 (44.4)	607 (47.5)	
Both	125 (4.5)	32 (4.6)	93 (4.5)		75 (4.4)	13 (3.1)	62 (4.9)	
**Tumor location index**				0.831				0.613
Single	2655 (95.5)	663 (95.4)	1992 (95.5)		1629 (95.6)	413 (96.9)	1216 (95.1)	
Bilateral and synchronous	83 (3.0)	24 (3.5)	59 (2.8)		47 (2.8)	9 (2.1)	38 (3.0)	
Bilateral and metachronous	42 (1.5)	8 (1.2)	34 (1.6)		28 (1.6)	4 (0.9)	24 (1.9)	
Bilateral				0.764				0.348
No	2688 (96.7)	675 (97.1)	2013 (96.5)		1645 (96.5)	416 (97.7)	1229 (96.2)	
Yes	92 (3.3)	20 (2.9)	72 (3.5)		59 (3.5)	10 (2.3)	49 (3.8)	
**Occurrence in other organs**				<0.001	-	-	-	-
No	2348 (84.5)	426 (61.3)	1922 (92.2)		-	-	-	-
Yes	432 (15.5)	269 (38.7)	163 (7.8)		-	-	-	-
**Histologic grade**				<0.001				<0.001
1	241 (8.7)	32 (4.6)	209 (10.0)		153 (9.0)	24 (5.6)	129 (10.1)	
2	1282 (46.1)	246 (35.4)	1036 (49.7)		786 (46.1)	153 (35.9)	633 (49.5)	
3	1257 (45.2)	417 (60.0)	840 (40.3)		765 (44.9)	249 (58.5)	516 (40.4)	
**p53(%)**	29.9 ± 24.7	33.4 ± 26.5	28.8 ± 24.0	<0.001	29.8 ± 24.3	33.1 ± 26.2	28.7 ± 23.5	<0.05
**Ki-67(%)**	27.3 ± 22.5	33.1 ± 26.5	25.3 ± 20.6	<0.001	28.1 ± 23.0	32.7 ± 26.3	26.6 ± 21.5	<0.001
**T stage**				<0.001				<0.001
0	134 (4.8)	17 (2.4)	117 (5.6)		70 (4.1)	11 (2.6)	59 (4.6)	
1	1569 (56.4)	278 (40.0)	1291 (61.9)		987 (57.9)	166 (39.0)	821 (64.2)	
2	950 (34.2)	316 (45.5)	634 (30.4)		567 (33.3)	196 (46.0)	371 (29.0)	
3	95 (3.4)	58 (8.3)	37 (1.8)		60 (3.5)	33 (7.7)	27 (2.1)	
4	32 (1.2)	26 (3.7)	6 (0.3)		20 (1.2)	20 (4.7)	0 (0.0)	
**N stage**				<0.001				<0.001
0	1862 (67.0)	315 (45.3)	1547 (74.2)		1140 (66.9)	199 (46.7)	941 (73.6)	
1	641 (23.1)	213 (30.6)	428 (20.5)		410 (24.1)	131 (30.8)	279 (21.8)	
2	200 (7.2)	110 (15.8)	90 (4.3)		105 (6.2)	61 (14.3)	44 (3.4)	
3	77 (2.8)	57 (8.2)	20 (1.0)		49 (2.9)	35 (8.2)	14 (1.1)	
**Tumor subtype**				<0.05				<0.001
Luminal A	1850 (66.5)	438 (63.0)	1412 (67.7)		1136 (66.7)	260 (61.0)	876 (68.5)	
Luminal B	363 (13.1)	96 (13.8)	267 (12.8)		205 (12.0)	52 (12.2)	153 (12.0)	
Basal	347 (12.5)	112 (16.1)	235 (11.3)		213 (12.5)	83 (19.5)	130 (10.2)	
HER2 overexpressing	220 (7.9)	49 (7.1)	171 (8.2)		150 (8.8)	31 (7.3)	119 (9.3)	
**Radiation treatment for curative**				<0.001				<0.001
No	724 (26.0)	267 (38.4)	457 (21.9)		479 (28.1)	178 (41.8)	301 (23.6)	
Yes	2056 (74.0)	428 (61.6)	1628 (78.1)		1225 (71.9)	248 (58.2)	977 (76.4)	
**Chemotherapy**				<0.001				<0.001
None	211 (7.6)	45 (6.5)	166 (8.0)		146 (8.6)	39 (9.2)	107 (8.4)	
Adjuvant	2056 (74.0)	412 (59.3)	1644 (78.8)		1266 (74.3)	260 (61.0)	1006 (78.7)	
Neoadjuvant	513 (17.1)	238 (34.2)	275 (13.2)		292 (17.2)	127 (29.8)	165 (12.9)	
**Anti-HER2**				<0.001				<0.001
No	2253 (81.0)	479 (68.9)	1774 (85.1)		1440 (84.5)	334 (78.4)	1106 (86.5)	
Yes	527 (19.0)	216 (31.1)	311 (14.9)		264 (15.5)	92 (21.6)	172 (13.5)	
**Anti-hormone**				<0.001				<0.001
No	656 (23.6)	210 (30.2)	446 (21.4)		416 (24.4)	150 (35.2)	266 (20.8)	
Yes	2124 (76.4)	485 (69.8)	1639 (78.6)		1288 (75.6)	276 (64.8)	1012 (79.2)	

BMI, body mass index; WBC, white blood cell; BCS, breast-conserving surgery; HER2, human epidermal growth factor receptor 2.

**Table 2 cancers-16-03799-t002:** Results of univariate and multivariate analyses.

	Overall Breast Cancer Group				Breast-Cancer-Only Group			
	Univariable		Multivariable		Univariable		Multivariable	
	HR (95% CI)	*p*-Value	HR (95% CI)	*p*-Value	HR (95% CI)	*p*-Value	HR (95% CI)	*p*-Value
**Age at diagnosis**	1.03 [1.03, 1.04]	<0.001	1.03 [1.02, 1.04]	<0.001	1.05 [1.04, 1.06]	<0.001	1.04 [1.03, 1.05]	<0.001
**Height**	0.96 [0.95, 0.97]	<0.001	0.99 [0.97, 1.00]	0.056	0.94 [0.92, 0.96]	<0.001	1.00 [0.98, 1.02]	0.785
**BMI**	1.04 [1.02, 1.06]	<0.001	0.98 [0.96, 1.01]	0.138	1.05 [1.02, 1.08]	0.002	1.01 [0.98, 1.03]	0.717
**Smoking**								
No	-		-		-			
Yes	1.38 [1.05, 1.81]	0.019	1.36 [1.02, 1.82]	0.037	1.25 [0.86, 1.82]	0.251		
**Drinking**								
No	-		-		-		-	
Yes	0.63 [0.51, 0.78]	<0.001	0.77 [0.61, 0.97]	0.027	0.52 [0.39, 0.69]	<0.001	0.52 [0.39, 0.69]	<0.001
**Age at menarche**	1.08 [1.03, 1.13]	0.001	0.96 [0.91, 1.02]	0.168	1.17 [1.09, 1.26]	<0.001	1.08 [1.01, 1.14]	0.017
**Age at menopause**	0.99 [0.97, 1.01]	0.387			0.98 [0.95, 1.01]	0.287		
**Parturition experience**								
No	-				-			
Yes	1.03 [0.82, 1.29]	0.801			1.12 [0.84 1.50]	0.426		
**Experience of oral contraceptives**								
No	-				-			
Yes	1.09 [0.86, 1.38]	0.492			1.05 [0.73, 1.49]	0.784		
**Hormone replacement therapy**								
No	-				-			
Yes	1.13 [0.86, 1.48]	0.367			0.86 [0.61, 1.22]	0.400		
**Family history**								
No	-				-			
Yes	0.73 [0.50, 1.06]	0.101			0.44 [0.25, 0.78]	0.005	0.60 [0.33, 1.09]	0.095
**Parents’ cancer history**								
Paternity	-				-		-	
Maternal line	0.58 [0.32, 1.03]	0.065			0.45 [0.22, 0.95]	0.037	0.61 [0.28, 1.31]	0.202
Parental	0.70 [0.30, 1.64]	0.408			0.66 [0.20, 2.19]	0.495	0.33 [0.00, 1.15]	0.082
None	1.26 [0.92, 1.74]	0.154			1.25 [0.82, 1.90]	0.309	1.07 [0.69, 1.65]	0.776
**Cancer history**								
No	-				-			
Yes	1.35 [0.93, 1.98]	0.118			1.11 [0.61, 2.01]	0.743		
**Total cholesterol**	1.00 [1.00, 1.00]	0.471			1.00 [1.00, 1.00]	0.092		
**Fasting glucose**	1.01 [1.00, 1.01]	<0.001	1.00 [1.00, 1.00]	0.005	1.00 [1.00, 1.01]	<0.001	1.00 [1.00, 1.01]	0.140
**WBC**	1.09 [1.06, 1.12]	<0.001	1.04 [1.00, 1.07]	0.028	1.07 [1.03, 1.11]	<0.001	1.01 [0.97, 1.05]	0.691
**Surgical type**								
None	-				-			
BCS	0.90 [0.47, 1.75]	0.764	0.62 [0.29, 1.32]	0.212	1.34 [0.56, 3.25]	0.513	1.34 [0.56, 3.25]	0.513
Mastectomy	1.99 [1.02, 3.89]	0.045	0.53 [0.24, 1.15]	0.108	2.74 [1.12, 6.73]	0.028	2.74 [1.12, 6.73]	0.028
**Tumor location**								
Left	-				-			
Right	0.94 [0.81, 1.09]	0.425			0.89 [0.74, 1.08]	0.251	0.62 [0.23, 1.65]	0.340
Both	0.93 [0.65, 1.33]	0.685			0.61 [0.35, 1.07]	0.086	0.49 [0.18, 1.32]	0.158
**Tumor location index**								
Single	-				-			
Bilateral & synchronous	1.26 [0.84, 1.89]	0.268			0.87 [0.45, 1.69]	0.687		
Bilateral & metachronous	0.55 [0.28, 1.11]	0.098			0.41 [0.15, 1.09]	0.075		
**Bilateral**								
No	-				-			
Yes	0.88 [0.57, 1.38]	0.583			0.70 [0.38, 1.32]	0.273		
**Occurrence in other organs**								
No	-		-					
Yes	3.84 [3.30, 4.48]	<0.001	2.57 [2.17, 3.04]	<0.001				
**Histologic grade**								
1	-		-		-		-	
2	1.49 [1.03, 2.16]	0.033	0.89 [0.59, 1.34]	0.581	1.37 [0.89, 2.11]	0.152	0.98 [0.62, 1.56]	0.936
3	2.79 [1.95, 3.99]	<0.001	1.14 [0.75, 1.74]	0.527	2.45 [1.61, 3.73]	<0.001	1.00 [0.61, 1.63]	0.991
**p53(%)**	1.01 [1.00, 1.01]	<0.001	1.00 [0.99, 1.00]	0.152	1.01 [1.00, 1.01]	<0.001	1.00 [1.00, 1.00]	0.825
**Ki-67(%)**	1.02 [1.02, 1.02]	<0.001	1.01 [1.01, 1.02]	<0.001	1.02 [1.01, 1.02]	<0.001	1.01 [1.00, 1.01]	<0.001
**T stage**								
0	-		-		-		-	
1	1.57 [0.96, 2.56]	0.072	1.28 [0.75, 2.17]	0.363	1.02 [0.56, 1.89]	0.937	0.88 [0.46, 1.68]	0.699
2	2.92 [1.79, 4.75]	<0.001	1.62 [0.94, 2.77]	0.082	2.15 [1.17, 3.95]	0.013	1.20 [0.62, 2.33]	0.596
3	6.52 [3.80, 11.20]	<0.001	2.38 [1.30, 4.34]	0.005	4.08 [2.06, 8.08]	<0.001	1.53 [0.71, 3.27]	0.276
4	15.89 [8.60, 29.33]	<0.001	2.65 [1.34, 5.22]	0.005	18.15 [8.66, 38.03]	<0.001	2.85 [1.25, 6.53]	0.013
**N stage**								
0	-		-		-		-	
1	2.05 [1.72, 2.44]	<0.001	1.47 [1.22, 1.78]	<0.001	2.07 [1.66, 2.59]	<0.001	1.64 [1.28, 2.09]	<0.001
2	3.50 [2.82, 4.35]	<0.001	1.99 [1.55, 2.56]	<0.001	3.90 [2.93, 5.19]	<0.001	2.83 [2.03, 3.96]	<0.001
3	6.07 [4.57, 8.05]	<0.001	3.25 [2.40, 4.40]	<0.001	4.90 [3.42, 7.02]	<0.001	2.89 [1.92, 4.37]	<0.001
**Tumor subtype**								
Luminal A	-		-		-		-	
Luminal B	1.09 [0.87, 1.35]	0.470	0.62 [0.48, 0.81]	<0.001	1.18 [0.87, 1.58]	0.286	0.95 [0.67, 1.35]	0.772
Basal	1.53 [1.25, 1.89]	<0.001	0.46 [0.33, 0.66]	<0.001	2.08 [1.62, 2.66]	<0.001	0.93 [0.60, 1.45]	0.745
HER2 overexpressing	1.05 [0.78, 1.41]	0.735	0.26 [0.17, 0.40]	<0.001	1.00 [0.69, 1.45]	0.989	0.52 [0.30, 0.91]	0.021
**Radiation treatment for curative**								
Yes	-				-		-	
No	1.89 [1.62, 2.20]	<0.001	1.80 [1.53, 2.13]	<0.001	2.03 [1.67, 2.46]	<0.001	2.21 [1.70, 2.64]	<0.001
**Chemotherapy**								
None	-				-			
Adjuvant	0.85 [0.63, 1.16]	0.315	0.62 [0.48, 0.81]	<0.001	0.70 [0.50, 0.98]	0.040	1.35 [0.87, 2.08]	0.177
Neoadjuvant	2.88 [1.62, 5.06]	<0.001	0.26 [0.17, 0.40]	<0.001	2.33 [1.62, 3.35]	<0.001	3.43 [2.10, 5.61]	<0.001
**Anti-HER2**								
No	-				-		-	
Yes	2.22 [1.89, 2.61]	<0.001	1.48 [1.19, 1.85]	<0.001	1.73 [1.37, 2.18]	<0.001	1.09 [0.80, 1.49]	0.566
**Anti-hormone**								
No	-				-		-	
Yes	0.60 [0.51, 0.71]	<0.001	0.34 [0.24, 0.47]	<0.001	0.49 [0.40, 0.60]	<0.001	0.50 [0.33, 0.76]	0.001

HR, hazard ratio; CI, confidence incidence; BMI, body mass index; WBC, white blood cell; BCS, breast-conserving surgery; HER2, Human epidermal growth factor receptor 2.

**Table 3 cancers-16-03799-t003:** Performance of the four machine learning algorithms.

		Accuracy	Precision	Recall	F1 Score	AUC	Specificity	Brier Score	MCC	AUPRC
Overall breast cancer group	*LR*	0.7950	0.5691	0.7410	0.6438	0.8467	0.8129	0.2050	0.5119	0.6662
	*RF*	0.8058	0.6566	0.4676	0.5462	0.8624	0.9185	0.1942	0.4370	0.6679
	*XGB*	0.8381	0.7634	0.5108	0.6121	0.8722	0.9472	0.1619	0.5314	0.7130
	*LGB*	0.8094	0.6093	0.6619	0.6345	0.8677	0.8585	0.1906	0.5066	0.7059
Breast-Cancer-only group	*LR*	0.7507	0.5000	0.7529	0.6009	0.8338	0.7500	0.2493	0.4493	0.6928
	*RF*	0.7859	0.6579	0.2941	0.4065	0.8043	0.9492	0.2141	0.3345	0.5648
	*XGB*	0.8504	0.8696	0.4706	0.6107	0.8518	0.9766	0.1496	0.5662	0.7013
	*LGB*	0.8270	0.6857	0.5647	0.6194	0.8482	0.9141	0.1730	0.5128	0.6676

AUC, area under the curve; MCC, Matthews correlation coefficient; AUPRC, area under the precision–recall curve; LR, logistic regression; RF, random forest; XGB, extreme gradient boosting; LGB, light gradient boosting machine.

## Data Availability

The data presented in this study are available on request from the corresponding author.

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
