# Peer review of "Mortality Prediction Modeling for Patients with Breast Cancer Based on Explainable Machine Learning"

_cancers, 2024, doi:10.3390/cancers16223799_

Round 1
Reviewer 1 Report
Comments and Suggestions for Authors
This manuscript describes the development of a machine-learning (ML) model to predict survival / death of South-Korean breast cancer patients from 31 clinical features. The manuscript is well written and organized. However, there are several methodological issues that need to be clarified.
(1) The supplementary material (figures, tables) was not available to this reviewer. It needs to be reviewed as well.
(2) Commonly in the literature, ML models are trained to predict 5-year survival of breast cancer patients. Here, the authors analyzed data from patients diagnosed with breast cancer between 2005 and 2020. From an independent source, they took info whether these patients had died until the end of 2021 or not. Hence, the cohort contains some patients that were diagnosed 16 years before the deadline and other patients that were diagnosed only 1 year ago. In my view, this may distort the interpretation of the results and needs to be explained and discussed.
(a) E.g. the authors found an age-effect, see table 1 695 dead patients had an average age of 54,7 years, survivors were on average 50.3 years old. If one was diagnosed in 2005, there would be a reasonable chance to die before 2021 anyhow, not necessarily due to breast cancer, but simply due to age. The chances are very different if one was diagnosed in 2020.
(3) Given that you analyzed an imbalanced data set, one has to be careful which performance metrics to use, see e.g. https://machinelearningmastery.com/tour-of-evaluation-metrics-for-imbalanced-classification/
Accuracy and specificity are no good measures in such cases, F1-score is a good measure on the other hand. Instead of AUC values, it is recommended to use Precision-Recall AUC curves. Please modify table 2 or explain why this problem does not apply in your case.
(4) Table 1 should list adjusted p-values that correct for multiple testing of the same cohort.
(5) Line 60: “Previous studies have mainly focused on considering only one risk factor …”. There are certainly many such studies, but there are now plenty of related ML studies that consider as many features as was done here. I suggest that you simply google for “machine learning breast cancer survival 2024”. This sentence should be reworded. You may also want to add further recent citations of related ML studies.
(6) Line 87: “were obtained through NLP of pathology reports.” Was that done by Statistics Korea, or in this manuscript?
(7) Line 146: “Imputation was conducted …”. Please list – separately for each clinical feature used here - the number of cases where this feature had to be imputed. The authors simply replaced missing numerical values by the mean of the available data points. In my opinion, KNN gives more reliable results, but this is what they did. How does the imputation of categorical variables “with their modes” work? This should be explained more clearly.
(8) Does your testing data set contain cases with imputed data? If this is the case, you may want to mention this as a potential caveat of your study. Ideally, I would only use complete data sets for testing.
(9) Line 227 “the models were trained using a previously selected optimal feature set.” A citation to that previous work needs to be added.
(10)Line 266: “Positive correlation”: this sounds like higher N stage (for example) has a better outcome, which is certainly not the case. Same with line 272 “grade increases have a positive effect on survivors”. Unfortunately, Fig. S2 was not available for review.
Author Response
We deeply appreciate your thoughtful review. We have carefully revised the manuscript based on the reviewer's comments, and the response will be submitted in a separate file.

Reviewer 2 Report
Comments and Suggestions for Authors
This study we developed a predictive classification models for breast cancer mortality using ML models and used the SHapley Additive exPlanation (SHAP) values to help explain the resulting model. This is an interesting manuscript with a number of techniques used for the analysis. However, the supplementary material was not available to review so this review was only performed using the main manuscript.
Comments are below:
MATERIALS AND METHODS:
- The authors quote that 9121 patients were excluded due to 'insufficient data' but have not quantified the impact of this exclusion and if this exclusion potentially introduced bias to the study results. This requires further justification.
- The authors claim they used NLP of pathology reports to classify the pathology data but details of this analysis and the results of these classifications are required.
- The authors performed propensity score matching (PSM) to 1:3 ratio. Justification of this ratio is required, more explanation about the PSM methodology and metrics used, and results of this analysis is required.
- The imputation methodology for the feature missing values requires expansion and justification
- Further explanation and justification is required for the min-max normalization pre-processing step is needed
- The manuscript would benefit from a flowchart of the analytical stages
- Please indicate your testing of assumptions of your Kaplan-Meier (KM) analysis
- Please include the Brier score and MCC in your metrics
- The calculation of SHAP values has potential issues with correlated features as there is the assumption of independence between features (e.g. kernel SHAP). Can you please indicate how you tested for this prior to running the SHAP analysis?
RESULTS:
- Table 2 - Your results do not indicate that the XGB performs as well as the LGB model for sensitivity/recall and F1-score. This would probably be due to imbalance, but recall seems an important metric for these models to detect breast cancer mortality. In addition, the other metrics for the LGB model are close to XGB so results from this model should have also been compared to ensure consistency in importance variables and SHAP results.
- Figure 3A - Further explanation surrounding Height is needed as this was identified as the fifth most important variable
- Figure 3B - Further explanation as to the results for P53(%) is needed and WBC
DISCUSSION:
- Your discussion is well written but further discussion about the low sensitivity in your selected XGB model is needed.
Author Response

(The authors gave the same response as above.)

Reviewer 3 Report
Comments and Suggestions for Authors
The paper did not address the proposed analysis well enough:
-The abstract has unnecessary subtitles as Backround/Objectives., Methods:, Results:. etc as well as abbrevations as XGB, SHAP.
-The part about literature review in Introduction section deals with only 11 discussed references that is not enough number to clear clarify the situation in the field were belongs the considered problem. Research question is missing. Also, please highlight controversial and diverging hypotheses for conducted research . Finally, briefly mention the main aim of the paper and highlight the main conclusions.
-In submitted manuscript section 2. Materials and methods should has two subsections.The first subsection 2.1 Matherials should be about used materials for conducted research – in submitted manuscript , subsections 2.1, 2.2 and 2.3 in existing paper and the second subsection 2.2 Methods should be with subsections 2.4, 2.5 and 2.6 in submitted manuscript.
In section 4. Discussion it is missing a comparsion with other state of the art methods in the field were belongs the considered problem and a paragrapg about future work. What happenes in the case of unbalanced dataset?Are the proposed measures valid then?
Section 5. Conclusion section is missing with briefly mention the main contribution of this research also conclusion is missing one sentence about future work in the field were belongs the considered problem
-Figure 3 could be slightly enlarged and thus clearer.
- The number from total 29 references in the paper could not be enough for the publication of the paper in such an eminent journal as IJGI especially that only 11 references are in literature review part of Introduction section.
Author Response

(The authors gave the same response as above.)

Round 2
Reviewer 1 Report
Comments and Suggestions for Authors
The authors have appropriately replied to my points.
Author Response
The comments and suggestions for authors by reviewer1 is that we have appropriately replied to reviewew points. Thanks for your consider for our research results. We sincerely appreciate your detail comments.
Reviewer 2 Report
Comments and Suggestions for Authors
Please refer to attached PDF.

Adequate English but suggest review of final manuscript.
Author Response
Thank you for your detail comments for improvement our research results.
We submit modified manuscript and materials (figures, tables) by your comments on attached files.
Thanks.
